# Effects of Sun Withering Degree on Black Tea Quality Revealed via Non-Targeted Metabolomics

**DOI:** 10.3390/foods12122430

**Published:** 2023-06-20

**Authors:** Zhuanrong Wu, Yuanfang Jiao, Xinfeng Jiang, Chen Li, Weijiang Sun, Yuqiong Chen, Zhi Yu, Dejiang Ni

**Affiliations:** 1National Key Laboratory for Germplasm Innovation and Utilization of Horticultural Crops, Wuhan 430070, China; 2020wzr@webmail.hzau.edu.cn (Z.W.); 2019305120024@webmail.hzau.edu.cn (Y.J.); chenyq@mail.hzau.edu.cn (Y.C.); 2Jiangxi Sericulture and Tea Research Institute, Nanchang 330202, China; jiangxinyue003@163.com (X.J.); hanwuji1110@126.com (C.L.); 3College of Horticulture, Fujian Agriculture and Forestry University, Fuzhou 350002, China; swj8103@126.com; 4College of Horticulture and Forestry Sciences, Huazhong Agricultural University, Wuhan 430070, China

**Keywords:** black tea, sun withering, metabolomics, aroma

## Abstract

In this study, the effects of different sun withering degrees (75% (CK), 69% (S69), 66% (S66), 63% (S63), and 60% (S60) water content in the withered leaves) on black tea sensory quality were investigated by means of sensory evaluation plus metabolomics analysis. Sensory evaluation results showed higher sensory quality scores for the black tea in S69–S66, due to better freshness, sweeter taste, and a sweet and even floral and fruity aroma. Additionally, 65 non-volatile components were identified using Ultra Performance Liquid Chromatography-Quadrupole-Time of Flight-Mass Spectrometry (UPLC-Q-TOF/MS). Among them, the content increase of amino acids and theaflavins was found to promote the freshness and sweetness of black tea. The aroma of tea was analyzed using combined Solvent Assisted Flavor Evaporation-Gas Chromatography-Mass Spectrometry (SAFE-GC-MS) and Headspace-Solid Phase Micro Extract-Gas Chromatography-Mass Spectrometry (HS-SPME-GC-MS), and 180 volatiles were identified, including 38 variable importance in projection (VIP) > 1 (*p* < 0.05) and 25 Odor Activity Value (OAV) > 1 volatiles. Statistical analysis revealed 11 volatiles as potential major aroma differential metabolites in black tea with a different sun withering degree, such as volatile terpenoids (linalool, geraniol, (E)-citral, and β-myrcene), amino-acid-derived volatiles (benzeneethanol, benzeneacetaldehyde, and methyl salicylate), carotenoid-derived volatiles (jasmone and β-damascenone), and fatty-acid-derived volatiles ((Z)-3-hexen-1-ol and (E)-2-hexenal). Among them, volatile terpenoids and amino acid derived volatiles mainly contributed to the floral and fruity aroma quality of sun-withered black tea.

## 1. Introduction

Black tea is the most consumed tea in the world, and its popularity with consumers is closely related to its unique quality and rich health functions. Studies have shown that black tea has various health functions, such as antioxidant, hypoglycemic, and hypolipidemic functions [1], and these health functions are closely related to its biochemical components [2]. The main functional components of black tea are tea polyphenols, amino acids, theaflavins, etc. The content of biochemical components is closely related to its processing process, and the optimization of different processes was reported as the core factor affecting the content of biochemical components [3,4]. The black tea processing process mainly consists of withering, rolling (cutting), fermentation, and drying [5]. Withering is the first step of black tea processing, which is one of the key links in the formation of black tea quality [6]. Current research on the withering process has focused on the effects of its temperature [7], time [8], and degree [9] on black tea quality.

In major black-tea-producing countries such as China, India, and Sri Lanka, black tea withering methods include outdoor sun withering, indoor natural withering, and warm-air withering [10]. Production practice shows that sun withering can not only save energy, shorten withering time, and reduce processing costs but also improve the taste and aroma quality of tea, which can improve the taste sweetness and freshness and aroma sweetness and floral fragrance. As previously reported, sun withering could facilitate the increase of terpene aroma compounds and significantly enhance the floral and fruity aromas of black tea [10,11]. Terpene volatile compounds are mainly derived from the synthesis of mevalonate (MVA) and methylerythritol phosphate (MEP) pathways, and light is the main environmental factor affecting the accumulation of terpene metabolites [12]. The common and major terpene volatile compounds in tea mostly exhibit floral and fruity aromas, with a high content in linalool and its oxides, geraniol, neroli tertiary alcohol, etc. In addition, despite differences of varieties, black tea possesses common volatiles such as phenylethanol and phenylethanal with floral characteristics, 2-ethyl-5-methylpyrazine and 2-pentylfuran with sweet aroma characteristics, jasmonone and jasmine endol with fruit aroma characteristics, and leaf alcohol and green leaf aldehyde with grassy flavor characteristics [13,14,15].

In recent years, due to the advantage of simultaneously identifying the composition and content of multiple substances, metabolomics has been increasingly applied in the study of the chemical composition of plants and flavor chemistry of foods. In the study of tea flavor chemistry, liquid chromatography–mass spectrometry (LC-MS) is mostly used to determine the changes in non-volatile components in tea, while gas chromatography–mass spectrometry (GC-MS) is mostly used for detection of volatile components. However, so far, sun withering methods still have some key issues to be clarified: (1) Under what light and temperature conditions can sun withering be carried out? (2) Should the whole withering process be performed only via sun withering or in combination with natural withering after a certain extent of sun withering? Answers to these issues will help to improve the sun withering method, which may not only reduce tea production costs but also improve black tea quality. In this study, we aimed to investigate the relationship between different sun withering degrees plus natural indoor withering and the volatile and non-volatile components associated with flavor characteristics of black tea and identify a satisfactory sun withering degree for improving its flavor quality in practical production using sensory evaluation combined with the metabolomics methods UPLC-Q-TOF/MS, SAFE-GC-MS, and HS-SPME-GC-MS.

## 2. Materials and Methods

### 2.1. Test Materials

The tea plant variety was Echa 10. The fresh leaves with one bud and two or three leaves were harvested on 29 April 2021 (sunny day) at the experimental tea garden of Huazhong Agricultural University (Wuhan, China).

### 2.2. Main Reagents

L-glutamine, L-lysine, L-theanine, L-arginine, L-isoleucine, L-leucine, L-phenylalanine, L-tryptophan, L-aspartic acid, L-asparagine, L-glutamic acid, L-threonine, L-proline, L-valine, L-tyrosine, L-methionine, theaflavin (TF), theaflavin-3-gallate (TF-3-G), theaflavin-3′-gallate (TF-3′-G), theaflavine-3,3′-digallate (TFDG), catechin (C), catechin gallate (CG), gallocatechin (GC), gallocatechin gallate (GCG), epicatechin (EC), epicatechin gallate (ECG), epigallocatechin (EGC), epigallocatechin gallate (EGCG), theobromine, theophylline, caffeine, epiafzelechin, kaempferol, kaempferide, quercetin, myricetin, vitexin, isovitexin, quercitrin, astragaline, quercetin-7-O-α-L-rhamnoside, quercetin-7-O-β-D-glucopyranoside, hyperoside, myricetin 3-O-galactoside, isovitexin 2′-O-arabinoside, glucosyl-vitexin, quercetin-3-o-rutinose, quercetin-3-O-D-glucosyl-(1-2)-L-rhamnoside procyanidin B1, procyanidin B2, fumaric acid, α-ketoglutaric acid, shikimic acid, caffeic acid, chlorogenic acid, gallic acid, etofylline, and cyclohexanone were purchased from Shanghai Yuanye Biotechnology Co. (Shanghai, China). The purity and manufacturer information of the above substances are shown in Appendix A. Volatile bound glycosides (GBVs) standards (Z)-3-hexenyl β-D-glucoside, benzyl β-D-glucoside, 2-phenylethyl β-D-glucoside, geranyl β-D-glucoside, nerol β-D-glucoside, benzyl β-D-primeveroside, 2-phenylethyl β-D-primeveroside, geranyl β-D-primeveroside, and nerol β-D-primeveroside were synthesized by the National Sugar Engineering Technology Center of Shandong University, and their purity was ≥95%.

### 2.3. Black Tea Processing

The freshly picked tea leaves were mixed and divided into 6 equal portions and spread flatly on 6 bamboo sieves of about 1 m in diameter, with the leaves thickness of about 3.5 to 4 cm. In this experiment, five treatments (S72, S69, S66, S63, S60) were performed with different fresh levels of sun withering, using indoor withering as a control. For the control (CK) test, the whole process was conducted indoors without sunlight and withered to 60% water content. For the S72, S69, S66, and S63 treatments, the harvested leaves (75% water content) were first exposed to the sun for outdoor withering to 72%, 69%, 66%, and 63% water content, respectively, followed by indoor withering (the same as CK) to 60% water content. For the full sun withering treatment (S60) test, the fresh leaves were directly withered to 60% water content outdoors in the sun. After withering, the leaves from all the groups were rubbed and cut for 15–20 s, then sieved through 60 mesh and placed in a fermentation machine (6CFJ-100, Zhejiang Green Peak Machinery Co., Ltd. (Quzhou, China)) at a temperature of 32 °C and 96% humidity for 2 h. After fermentation, the leaves were first dried for 10 min using a 110 °C tea aromatizer, stretched out to cool to room temperature, then dried for 30 min under 90 °C, stretched out to cool, and then further dried under 80 °C for 1 h to finish the drying process.

### 2.4. Black Tea Sensory Quality Evaluation

Referring to the national tea evaluation standard [16], the sensory evaluation was performed by five professional tea evaluators, who scored brew color, aroma, taste, and infused leaves at the same time. The evaluators have obtained the national “senior tea assessor” qualification certificate.

### 2.5. Determination of Main Quality Components of Black Tea

Total tea polyphenol: folin-phenol reagent method [17]. Total free amino acid: ninhydrin colorimetric method [17]. Water soluble carbohydrate: anthrone-sulfuric acid colorimetric method [9]. Total theaflavins, thearubigins, and theabrownins: organic reagents method [18].

### 2.6. UPLC-Q-TOF/MS Analysis of Non-Volatile Components of Black Tea

Briefly, the sample was weighed 150 mg in a 20 mL volumetric flask, 7.5 mL of 75% (v/v) methanol solution was added, and 150 μL of 250 μg/mL etofylline was used as the internal standard. Extraction for 30 min in a 70 °C water bath. The supernatant was centrifuged (5000 r, 3 min) after cooling, and the supernatant was filtered through 0.22 μm and prepared for use [9]. UPLC-Q-TOF/MS was used as the liquid chromatography instrument, and Zorbax Eclipse Plus C18 (100 × 2.1 mm, 1.8 μm, Agilent, Santa Clara, CA, USA) was used as the column for chromatographic analysis. Mass spectrometry conditions were referred to our previous experiments [19].

### 2.7. Analysis of Volatile Components of BLACK Tea by SAFE-GC-MS and HS-SPME-GC-MS

Aroma substances were extracted by means of headspace solid-phase microextraction (HS-SPME) using a microextraction fiber coated with polydimethylsiloxane/divinylbenzene (PDMS/DVB, 65 µm, 1 cm) [20]. Before analysis, 1 g of black tea powder was weighed into a 20 mL headspace vial, followed by adding 5 mL of boiled saturated NaCl solution, adding 1 mL of cyclohexanone solution as internal standard (cyclohexanone concentration: 100 μL/L), and closing the vial immediately. The headspace flask was placed in a water bath at 60 °C for 1 h. Meanwhile, solvent-assisted flavor extraction (SAFE) was used to extract the aroma compounds. Briefly, 10 g of black tea, 500 mL of boiling ultrapure water, and 0.5 mL of cyclohexanone solution were added to a 1000 mL beaker, followed by maceration extraction for 10 min, filtering, cooling to room temperature, and setting the tea broth aside; the extraction was repeated five times with 60 mL of dichloromethane for 30 min. Then, the combined organic extracts were subjected to distillation in a high-vacuum (5 × 10^−3^ kPa) SAFE apparatus, followed by drying the obtained extracts with anhydrous sodium sulfate, and concentrating the extracts to about 10 mL via Wechsler distillation and then to 500 μL via nitrogen blow. Finally, 1 μL of the sample solution was injected into the gas chromatographic column using a 10 μL syringe for analysis. Mass spectrometry conditions followed our previous experiments [9]. The test was repeated three times for each sample.

### 2.8. Data Statistics and Analysis

Data are expressed as mean ± standard deviation (SD) of 3 independent experiments, and statistical analysis was performed with SPSS Statistics 26 software, using least significant difference (LSD) multiple comparisons, principal component analysis (PCA), and orthogonal partial least squares discriminant analysis (OPLS-DA). Significant difference between different groups was considered at *p* < 0.05. Correlation analysis was performed using Pearson correlation coefficient. Plots were performed using Origin 2022, Simca 14.1.

## 3. Results and Discussion

### 3.1. Effects of Different Sun Withering Degrees on the Appearance of Fresh Leaves

The authors’ research team has previously found that sun withering at a temperature of 20–32 °C and light intensity of 25,000 Lux in April–May could significantly improve the black tea quality (unpublished data). The suitable temperature for indoor withering is 25–35 °C and humidity < 60%, under which better quality black tea can be prepared [21]. The outdoor temperature and light intensity, indoor temperature and humidity, and withering time recorded during this experiment are shown in Appendix A, where the sun withering and indoor withering conditions were seen to meet the withering requirements of black tea in this experiment. Additionally, the use of sun withering was also shown to shorten the whole withering time and improve the production efficiency.

The changes of fresh leaves’ appearance during sun withering are shown in Figure 1. As the sun withering degree gradually increased, the bright green fresh leaves gradually lost luster and turned dark green while showing different degrees of quality changes. With the increase of sun withering degree, leaf damage could be easily observed locally with the appearance of red color. This phenomenon was also observed in oolong tea after 60 min and 120 min of sun withering, where the leaves had severe localized red-browning and the top leaves were extensively burned and embrittled [11]. If the sun withering degree was controlled at S66, the red edge of the leaves was not obvious, inferring that the S66 may be the basic standard sun withering degree, and excessive withering may affect the quality of tea.

### 3.2. Effects of Different Sun Withering Degrees on the Sensory Quality of Black Tea

As shown in Figure 2A, the degree of sun withering had a greater effect on the taste and aroma quality of black tea. Based on the total evaluation score, the sensory score increased and then decreased with the gradual increase of sun withering degree, reaching the highest in S69, followed by a gradual decrease, and the sensory score was higher in S60 than in CK. In terms of aroma quality, as the degree of sun withering gradually increased, the green aroma weakened, and the aroma gradually became pure with sun withering lower than S69; a sweet and even floral and fruity aroma appeared (Figure 2B). In Figure 2A, the aroma scores were significantly higher in the sun-withered black teas relative to the control, with the peak value in S69, indicating that sun withering could significantly improve the aroma quality of black tea. This phenomenon was also observed in Keemun black tea and oolong tea after sun withering, which enhanced the floral and fruity aromas of Keemun black tea and the clear and light floral and fruity aromas of oolong tea [10,12]. In terms of taste quality, with the gradual increase of sun withering, the greenish astringency decreased, the freshness and sweetness increased, and the taste gradually became sweet and mellow with sun withering below S69 (Figure 2C). The taste scores were significantly higher for sun-withered black teas versus the control, with the peak value in the S69 sample, indicating that sun withering could also significantly improve the taste quality of black tea. This phenomenon was also observed in oolong tea processing, where tea made from oolong tea after 45 min of sun withering showed a reduction in astringency and an increase in taste mellowness in the tea broth [12].

### 3.3. Effects of Different Sun Withering Degrees on the Main Quality Components of Tea Leaves

#### 3.3.1. Effects of Different Sun Withering Degrees on Non-Volatile Components of Black Tea

A total of 65 compounds were identified in the experiment (Appendix A), including 16 amino acids, 8 catechins, 4 theaflavins, 19 flavonoids, 3 alkaloids, 6 organic acids, and 9 GVBs, with a total of seven classes, and their relative contents are shown in Figure 3B and Appendix A. The PCA results showed that the non-volatile substances in tea leaves gradually changed during sun withering (Figure 3A), and compared with CK, the sun withering treatment groups changed significantly in substance, especially the S60 sample, which had a large difference from the other treatments.

Amino acids are one of the main contributors to the taste and aroma quality of tea. Compared with CK, the total amount of amino acids in the tea showed a decreasing trend in the sun withering treatment groups versus CK, indicating that sun withering treatment may have promoted amino acid conversion during processing [22]. At the same time, the amino acid species showed a different change trend, with an uptrend in L-glutamine, L-glutamic acid, L-methionine, L-arginine, L-theanine and L-proline; a downtrend in L-phenylalanine, L-tryptophan, L-tyrosine, L-leucine, L-isoleucine, L-threonine and L-asparagine; and an uptrend followed by a downtrend for L-valine. Amino acids were further classified according to umami, sweet, and bitter tastes [9,23], and the variation of amino acid content in each category for different degrees of sun withering is shown in Figure 4A; the ratio of each category was calculated (Figure 4B). Compared with CK, the sun withering treatment groups showed an increase of nearly 20% in the percentage of umami amino acids and a decrease of about 17% in the percentage of bitter amino acids, indicating that the amino acid categories changed after sun withering, which could significantly improve the taste quality of tea [23].

Among the amino acids with an increased content, L-theanine is the most abundant amino acid in tea (about 50–60% of the total amino acids) with sweetness and freshness characteristics, and its content increase contributes to the fresh taste [19]. The correlation analysis showed a positive correlation between L-theanine and taste sensory score. During withering, L-theanine tended to change in two aspects. On the one hand, although in an isolated state after picking, the fresh leaves still had strong physiological functions. Sunlight exposure and temperature were reported to affect the accumulation of L-theanine, with an increase in its accumulation under a suitable temperature and short-time exposure, leading to its increase after sun withering, indicating that sunlight can promote the synthesis of L-theanine in isolated fresh leaves [24,25]. This can be supported by the study of [26], who found that yellow light withering could increase the content of black tea L-theanine under the same withering time relative to the dark treatment. On the other hand, the withering process in this experiment consisted of two procedures: sun withering for a longer time and indoor withering for a short time. L-theanine content has been reported to decrease with increasing withering time [27]. The total withering time decreased with the increase of sun withering time. This suggests that a longer sun withering time of black tea may have led to L-theanine degradation. L-arginine is also a main component of the freshness of tea broth, and its content is second only to L-theanine; its change pattern is consistent with that of L-theanine.

Studies have shown that free amino acids in tea leaves can be partially converted into volatile compounds. On the one hand, during the drying stage of black tea, amino acids underwent a Maillard reaction, enabling sugar substances to form volatile compounds such as pyrazines and pyrroles that contribute to the sweet aroma of black tea. On the other hand, under the action of enzymes, amino acids can undergo decarboxylation and deamination reactions to form aromatic compounds such as alcohols, aldehydes, and indoles [28]. L-phenylalanine, L-tryptophan, and L-tyrosine belong to aromatic amino acids. Among them, L-phenylalanine and L-tyrosine can generate aroma products such as benzeneethanol, benzeneacetaldehyde, and methyl salicylate under the action of various enzymes such as amino acid transaminase [19,29], while L-tryptophan is a precursor substance for the synthesis of the floral fragrance compound Indole [30]. Compared to indoor withering, sun withering showed a decrease of 25–35%, 72–85%, and 5–10% in the content of L-tryptophan, L-phenylalanine, and L-tyrosine, respectively, as shown in Appendix A. This was also consistent with our aroma study results, where the contents of benzyl alcohol and methyl salicylate increased by 1~2% and 14~27%, respectively, under sun withering treatment relative to indoor withering. Moreover, indole was not detected in the indoor withering treatment but formed in large amounts under sun withering treatment, reaching 0.795 µg/g to 1.490 µg/g. Benzyl alcohol and methyl salicylate are floral aroma compounds, and their content increase could improve the aroma quality of the finished tea [31]. One possible explanation is the greater effect of sun withering’s high temperature adversity stress on aromatic amino acids, leading to their further conversion into aroma substances [32]. Another possible explanation is the sunlight irradiation-induced expression of certain aroma substance synthesis genes in tea leaves [33], a phenomenon that has been demonstrated in strawberry [34], melon [35], and other postharvest fruits and vegetables in the synthesis and transformation of flavor substances. Additionally, decreased levels of L-leucine and L-isoleucine are known to contribute to the formation of volatiles [29]. L-isoleucine is considered a key precursor of many amino acid branched volatiles, but its conversion to volatiles has not been clarified [29]. L-leucine is involved in the synthesis of carotenoids, while carotenoid degradation can form floral aroma compounds such as jasmone, neryl acetone, α-ionone, and β-ionone, which could improve the aroma quality of black tea. β-ionone is the main degradation product of β-carotene. Moreover, our aroma study results revealed a significantly higher content of jasmone and neryl acetone after sun withering versus indoor withering, with the highest content in S69, and an increase of 21% and 52%, respectively. However, the content of β-ionone decreased, probably because β-ionone not only underwent degradation reactions but also photo- and auto-oxidation reactions under sunlight conditions [36].

Sun withering had a greater effect on tea polyphenols. Compared with CK, sun withering treatment produced black tea with significantly higher polyphenol content, with the highest increase of 12% (Appendix A). The main components of tea polyphenols are catechins, and eight catechins were identified in the experiment, including four non-ester-type catechins (C, EC, GC, EGC) and four ester-type catechins (CG, ECG, GCG, EGCG). The total amount of catechins retained in black tea showed a trend of increasing first and then decreasing with increasing sun withering degree. Interestingly, the total retention of both ester-type catechins and non-ester-type catechins also showed a trend of increasing first and then decreasing. Among them, the differences in ester-type catechin content were mainly manifested by the EGCG and EGC content. The total amount of ester-type catechins for each treatment was CK: 0.520 mg/g, S72: 1.011 mg/g, S69: 1.214 mg/g, S66: 0.801 mg/g, S63: 0.683 mg/g, and S60: 0.408 mg/g. The total amount of non-ester-type catechins was CK: 0.329 mg/g, S72: 0.677 mg/g, S69: 0.837 mg/g, S66: 0.572 mg/g, S63: 0.459 mg/g, and S60: 0.648 mg/g. During black tea processing, catechins are oxidized to form theaflavins, thearubigins, and theaflavins [28]. Therefore, the catechin content retained in the tea can reflect its content during the withering process. As previously reported [37,38], the genes involved in the synthesis of catechin derivatives were susceptible to light, and shade treatment could reduce the content of most catechins in preharvest tea leaves. In another study [18], the relative total amount of catechins was shown to be significantly higher in summer and autumn black tea processed with red light irradiation withering than in the control, mainly in the content difference of EGC and GCG. Light has also been shown to promote the increase of POD and PPO enzyme activities, which can facilitate the oxidation of catechins during subsequent kneading and fermentation [19]. The retention of catechins was significantly lower in S60 than in CK, indicating that the oxidative transformation of catechins is also accelerated by a long sun withering time.

Theaflavins are oxidation products of catechins, and the four analyzed theaflavin contents showed similar change trends. Compared with CK, sun withering significantly increased the content and total amount of each fraction, with the highest value for S69, probably due to the increase in catechin content from sun withering, resulting in an increase in its conversion to theaflavin. Theaflavins in black tea could affect the concentration, strength, and freshness of tea broth, and their content increase is beneficial for improving black tea quality [28]. Correlation analysis showed that all four theaflavin fractions were highly significantly and positively correlated with taste score and total sensory score, with correlation coefficients higher than 0.7 for TF, TF-3-G, and TF-3′-G (Appendix A), further indicating that sun withering could improve fresh taste mainly through increasing theaflavin content.

With the increase of sun withering degree, the total amount of GVBs showed a trend of increasing first and then remaining relatively stable in the sun withering range of S69–S60. It has been reported that red light irradiation at the withering stage of black tea could increase the activity of glycosidases, thus affecting the synthesis and accumulation of GBVs [20]. With the increase in sun withering degree, the flavonoid content first showed an uptrend and then a downtrend, reaching a peak at S66, while the caffeine content remained basically unchanged, indicating that sun withering had little effect on caffeine. Furthermore, six organic acids were detected, mainly including shikimic acid, α-ketoglutaric acid, and caffeic acid, with the highest content for shikimic acid (8.700 mg/g–9.548 mg/g).

#### 3.3.2. Effects of Different Sun Withering Degrees on Volatile Components of Black Tea

Sensory evaluation indicated that as the sun withering degree gradually increased, the tea green aroma weakened, especially the S69 sun-withered tea, which exhibited a sweet and even floral and fruity aroma. The effects of different sun withering degrees on the aroma quality of black tea were further analyzed using both SAFE-GC-MS and SPME-GC-MS to explore the changes of volatile components of black tea. It has been reported that the extraction of tea volatiles via SAFE could better preserve the aroma of the samples to provide more complete volatile information [39]. The SAFE results showed that 269 aroma substances were isolated from CK, and the number of aroma substances was significantly increased under sun withering treatment. Compared to CK, the S72, S69, S66, S63, and S60 treatments showed an increase of 23, 21, 29, 29, and 35 aroma substances, respectively, suggesting the formation of a large number of aroma substances during sun withering, which may further promote the formation of aroma substances in the subsequent fermentation process. This also suggests that the increase in sun withering degree may favor the formation of aroma substances. After identification according to standard RI value and MS, 180 aroma substances were detected via both SAFE-GC-MS and SPME-GC-MS (Appendix A), including 50 alcohols, 30 aldehydes, 15 ketones, 36 acids and esters, 38 hydrocarbons, and 11 others. In Figure 5B, alcohols were seen to cover the largest proportion, accounting for over 60% of the total volatile components. The relative contents of alcohols, ketones, acids and esters, hydrocarbons, and other substances showed an uptrend first and then a downtrend as the sun withering degree gradually increased, reaching the peak in S69–S66, while the relative content of aldehydes showed a decreasing trend, with the highest content in CK. Meanwhile, as shown in Figure 5A, the total aroma of black tea varied significantly with sun withering degrees, with the highest total aroma in S69, which was significantly different from CK. In addition, the total aroma was also higher in S66 and S63 than in CK, but significantly lower in S60 than in CK, indicating that excessive sun withering would affect or even reduce the aroma quality. Further analysis revealed that the proportion increased in alcohols, acids, and esters, decreased in aldehydes and ketones, and was more stable in hydrocarbons, etc. in sun-withered tea relative to indoor-withered tea (CK) (Figure 5B). This indicates that the changes in these aroma categories may cause differences in the aroma quality of tea with different sun withering degrees [10].

To gain insight into the changes in aroma quality of black tea with different sun withering degrees, multivariate statistical analysis was conducted. PCA showed significant differences in the aroma composition of black tea in different treatment groups (Figure 6B), especially CK and S60, which were far away from other treatment points, indicating some differences in the aroma quality of black teas processed with a different sun withering degree, and the non-sun-withered and fully sun-withered black teas showed larger aroma differences from the other sun-withered black teas. Additionally, 38 differential metabolites with variable importance in projection (VIP) > 1 and *p* < 0.05 were screened using OPLS-DA analysis, and the content variation in each substance is shown in Figure 6A. Odor activity value (OAV) is widely used to assess the contribution of aroma compounds to aroma quality [40]. In this study, we also calculated the OAV value for each substance based on the odor thresholds reported in the literature and identified 25 aroma substances with OAV > 1, which are considered to contribute significantly to the overall aroma profile. They include alcohols (linalool, benzeneethanol, nerol, (Z)-hex-3-en-1-ol, geraniol, 1-octene-3-ol), aldehydes ((E)-2-hexenal, octanal, benzeneacetaldehyde, (2E,6Z)-nona-2,6-dienal, (2E)-2-nonenal, decanal, β-cyclocitral, (2E)-2-decenal, (E)-citral, (2E)-2-octenal)), ketones (β-damascenone, jasmone, β-ionone), acids and esters (hexanoic acid, methyl salicylate, dihydroactinidiolide, δ -decalactone), and others (β-myrcene, 2,6,10,10-tetramethyl-1-oxaspiro [4.5]dec-6-ene); OAV data are shown in Table 1. Substances with an OAV value greater than 100 were linalool, geraniol, (E)-2-hexenal, benzeneacetaldehyde, (2E,6Z)-nona-2,6-dienal, decanal, β-damascenone, jasmone, dihydroactinidiolide, and δ-decalactone; the OAV value of β-damascenone exceeded 10,000.

The potential differential metabolites in different sun withering treatments were explored using a combined analysis of OPLS-DA (VIP > 1; *p* < 0.05) and OAV (OAV > 1), and the results are shown in Figure 6C. A total of 11 differential metabolites were screened, namely linalool (OAV: 404–504), geraniol (OAV: 426–488), benzeneethanol (OAV: 1–2), (Z)-3-hexen-1-ol (OAV: 4–10), benzeneacetaldehyde (OAV: 3120–5027), (E)-2-hexenal (OAV: 264–308), (E)-citral (OAV: 4–7), jasmone (OAV: 130–182), β-damascenone (OAV: 18,714–28,318), methyl salicylate (OAV: 19–26), and β-myrcene (OAV: 1–2). The 11 volatiles can be considered as the potential major aroma differential metabolites in black teas with a different degree of sun withering. They are mainly derived from amino-acid-derived volatiles (AADVs), volatile terpenoids (VTs), fatty-acid-derived volatiles (FADVs), and carotenoid-derived volatiles (CDVs), and their content varies as shown in Appendix A–D.

Alcohols are the key volatiles for the aroma quality of black tea, and most of them have floral and fruity aroma or sweet aroma characteristics. With the gradual increase of sun withering, the total amount of alcohols showed a trend of rising first and then falling, with the largest amount in S69. As shown in Figure 6A, the VIP > 1 (*p* < 0.05) alcohol differential metabolites were mainly terpenoids such as linalool, its oxides, dehydrolinalool, geraniol, and nerolidol with floral and fruity aroma characteristics, accounting for about 80% of the total alcohols in the different sun-withered dried teas and playing an important role in the formation of aroma quality. As the degree of sun withering gradually increased, the content of these alcohols showed a decreasing trend, except for the content of nerolidol with an uptrend first and then a downtrend. The highest content was observed in S69–S66, which may affect the floral and fruity aroma of black tea. Studies have reported light as the main environmental factor affecting volatile terpenoid accumulation and the expression patterns of numerous genes involved in terpenoid metabolic pathways [12]. Among them, sun withering promoted the up-regulated expression of genes related to DXS, HDS, and CMK in the methylerythritol phosphate pathway (MEP) for terpenoid synthesis, thus enhancing terpenoid metabolism [12], and light intensity is highly correlated with the accumulation of monoterpenes and sesquiterpenes [50]. In addition, alcohols exist not only in free form but also as glycosides, and during black tea processing, volatiles bound to glycosides (linalool and its oxides, benzeneethanol, geraniol, etc.) are hydrolyzed and actively participate in the regulation of aroma [51], probably because during sun withering, light and heat not only promote the activity of endogenous hydrolytic enzymes but also the expression of genes related to some small-molecule volatile compounds (e.g., terpenoids) [10,52]. Therefore, the significant increase in terpenoids after sun withering might improve the floral and fruity aroma quality of black tea. However, under excessive sun withering, the expression of heat shock proteins and resistance genes in leaves was up-regulated, and their up-regulation could inhibit the expression of aromatic substance-synthesis genes [11]. In the present study, excessive sun withering (S60) caused a significant decrease in terpene content, a phenomenon also observed during oolong tea sun withering. This was consistent with a previous report that the degree of sun withering should not be too heavy; otherwise, it is also detrimental to black tea quality [11]. In addition, the SAFE test results showed a significant decrease in the content of (Z)-3-hexen-1-ol (a grass-flavored volatile with an important impact on tea quality) with the increase of sun withering degree compared to indoor withering, and the longer the sun withering time, the lower the (Z)-3-hexen-1-ol content, consistent with the sensory evaluation results. In the sensory evaluation, a green herbaceous flavor was heavy in CK, but decreased significantly in S72, also indicating that moderate sun withering is beneficial for reducing the green grass aroma of tea.

The aldehydes in tea generally present a grassy or clear aroma, and some of them have a floral character. The relative total amount of aldehydes showed a decreasing trend with a gradual increase in sun withering degree. Among them, the total amount of aldehydes did not differ significantly between treatments S72–S63 but decreased significantly (26%) in S60 relative to the other treatments, indicating that sun withering had a greater effect on aldehydes, and the longer the sun withering time, the lower their content. Most compounds with a grassy flavor (decanal, (2E)-2-nonenal, (2E)-2-decenal, and (E)-2-octenal) showed a decreasing trend with the gradual increase of sun withering degree [53]. (E)-2-hexenal and (2E,6Z)-nona-2,6-dienal, the main green grass substances, are fatty-acid-derived volatiles, mainly derived from the oxidation of fatty acids [10]. Compared to CK, sun withering treatments showed an increase in the content of (E)-2-hexenal, probably because the conditions of light and heat during sun withering enhanced the activity of lipoxygenase (LOX) and promoted fatty acid oxidation [10]. Interestingly, the SAFE assay results indicated that with a gradual increase of sun withering degree, (E)-citral with a fruity aroma showed an increasing trend, while 2-hexenal with a sweet fruity aroma showed an uptrend first and then a downtrend, significantly higher in S69–S66 than in the other treatments, which might be beneficial to the formation of the sweet aroma quality of black tea.

Ketones with high aroma activity and very low threshold are also important components in the formation of black tea’s characteristic aroma [42]. β-damascenone and β-ionone, contributors to floral and sweet aromas, showed an increasing trend with increasing sun withering degree, while jasmone showed a trend of increasing first and then decreasing, with higher contents in S69–S63 (Figure 6A). β-ionone and β-damascenone were reported to have floral and fruity aromas and are also products of carotenoid degradation, and sun withering may promote the conversion/degradation of carotenoids [10]. Additionally, SAFE detected 6-methylhept-5-en-2-one with a lemon aroma, geranylacetone with a sweet aroma, and beta-dihydro-ionone with a floral aroma, with the content of geranylacetone being significantly increased in S69–S66, which might contribute to improve sweet floral aroma formation in black tea.

Acids and esters contributed to the aroma of black tea, and their content increased significantly after processing [42]. With the gradual increase in sun withering degree, the total amount of acids and esters showed a trend of increasing first and then decreasing, with their content higher in S69–S63 than in the other treatments. Among acid compounds, the contents of (E)-3-hexenoic acid and geranic acid were higher, and among esters, the contents of methyl salicylate, (Z)-3-hexenyl butyrate, and linalyl formate increased after sun withering, with methyl salicylate as the most abundant substance, accounting for about 50% of esters. Compared to CK, the content of methyl salicylate increased by 14%, 27%, and 17% in S69, S66, and S63 treatments, respectively, probably because the light and heat during sun withering promoted the conversion of the aromatic amino acid phenylalanine [10]. Additionally, the content of δ-decalactone with sweet aroma characteristics increased significantly in S69–S63 versus CK. Moreover, the SAFE results also revealed a significant content increase of δ-octalactone, dimethyl phthalate and dibutyl phthalate with sweet and floral fragrance characteristics in sun withering treatments relative to CK. Among them, dimethyl phthalate, a valine derivative with floral aroma characteristics, was also the main differential metabolite, and its content increased significantly in S69–S66, probably due to the conversion of valine promoted by the light and heat of sun withering.

Other compounds were also detected in this study, such as hydrocarbons, mainly including β-myrcene, α-farnesene, β-himachalene, 2,6,10,10-tetramethyl-1-oxaspiro [4.5] dec-6-ene, indole, etc. Among them, β-myrcene with a fatty odor had the highest content and was also a differential metabolite between treatments. Additionally, SAFE also detected hydrocarbons such as styrene, longleafene, β-pinene, limonene, and terpinolene with aromatic hydrocarbon characteristics as well as volatiles such as 2-methyltetrahydrofuran, 3,4-dihydro-2H-pyran, 3-methyl-4,5-dihydrofuran, and 1-ethylpyrrole-2-carbaldehyde with roasted aroma characteristics. These substances were higher in S69-S66 than in CK, indicating that sun withering was beneficial for improving aroma quality. Among them, the amino acid derivative 2-methyltetrahydrofuran and the roasted analog 1-ethylpyrrole-2-carbaldehyde were the main differential metabolites and significantly increased in content from S69 to S66. The decrease of 2-methyltetrahydrofuran content in S66 may be due to high-temperature leaf scorching during excessive sun withering, thus inhibiting its transformation. In contrast, 1-ethylpyrrole-2-carbaldehyde with roasted aroma characteristics increased significantly after sun withering, presumably due to the formation of such substances promoted by the continuous light and temperature during sun withering.

## 4. Conclusions

As show in Figure 7, This study demonstrated that sun withering degree has a great influence on black tea sensory quality. Sensory evaluation results revealed the highest sensory quality score in the sun withering range of S69–S66. Meanwhile, LC-MS and GC-MS results confirmed that the content of theaflavin and amino acid was also higher in the range of S69–S66, and so was the content of aroma substances with floral and fruity characteristics such as linalool and geraniol. Additionally, the formation of aroma substances under sun withering was found to be associated with the metabolic pathways of terpenoid biosynthesis, amino acid hydrolysis, carotenoid degradation, and fatty acid oxidation during withering, but the related mechanisms need to be further investigated. Overall, it can be concluded that moderate sun withering (to 69–66% water content) plus indoor withering (to 60% water content) contribute significantly to improve the aroma and taste quality of black tea.

## Figures and Tables

**Figure 1 foods-12-02430-f001:**
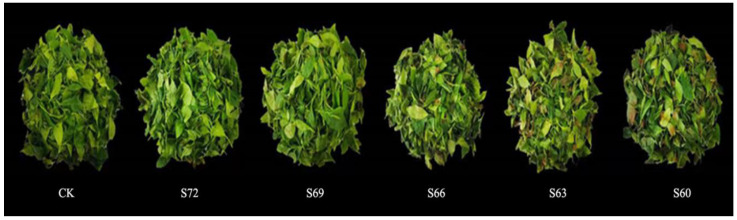
Changes in the appearance of fresh leaves with different degrees of sun withering.

**Figure 2 foods-12-02430-f002:**
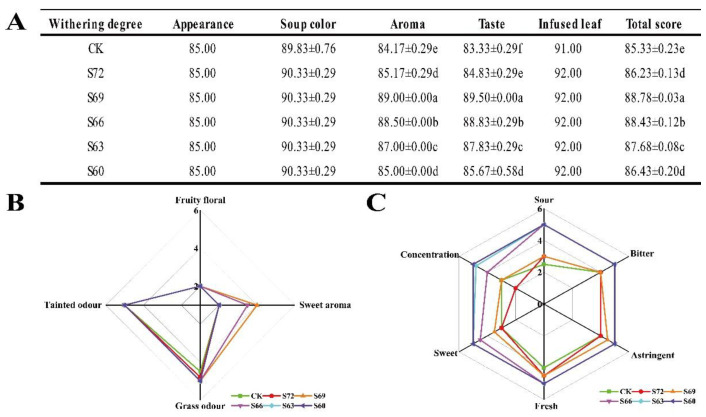
Sensory evaluation results of black tea with different sun withering degrees. (**A**) Sensory evaluation scores of black tea with different degrees of sun withering. (**B**) Radar chart of aroma factor scores of black tea with different degrees of sun withering. (**C**) Radar chart of taste factor scores of black tea with different degrees of sun withering. The sensory scores were evaluated by five experts, with total score including appearance ×20% + brew color × 10% + aroma × 30 + taste × 30 + infused leaf × 10. Different lowercase letters in the same column indicate significant differences at *p* < 0.05 level.

**Figure 3 foods-12-02430-f003:**
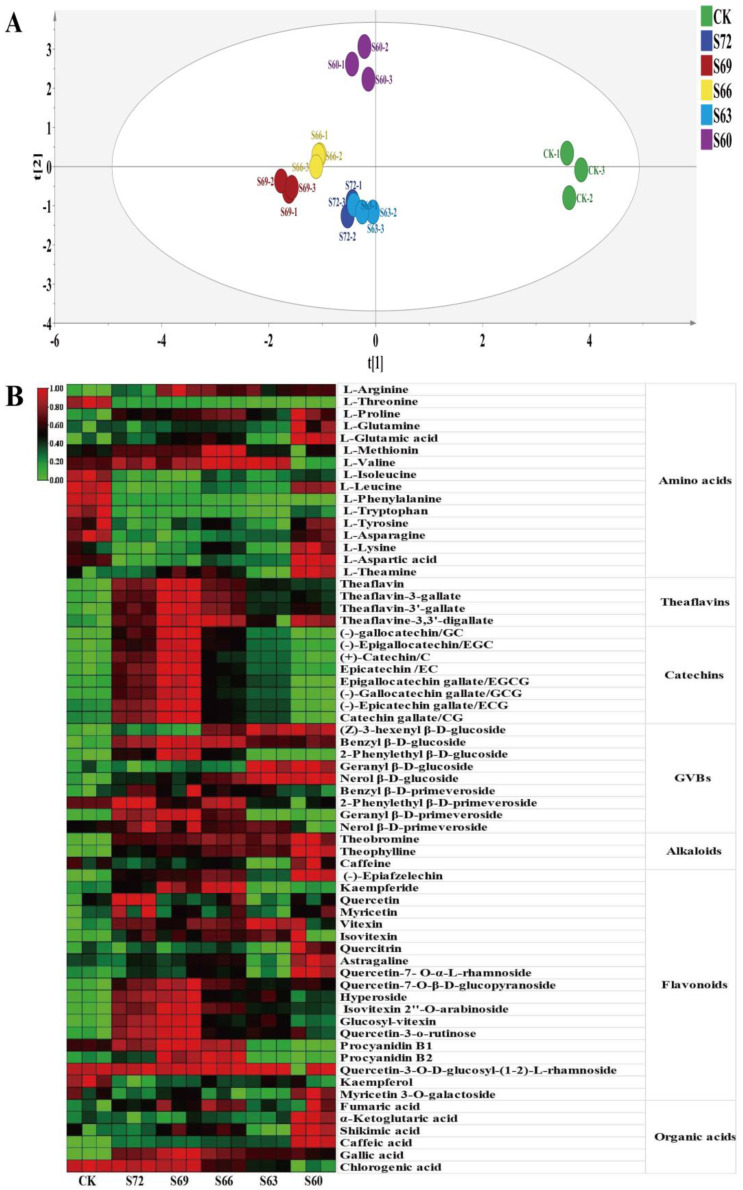
Analysis of non-volatile components of black tea with different degrees of sun withering. (**A**) PCA plot (R^2^ = 0.941, Q^2^ = 0.761). (**B**) Heat map of non-volatile components.

**Figure 4 foods-12-02430-f004:**
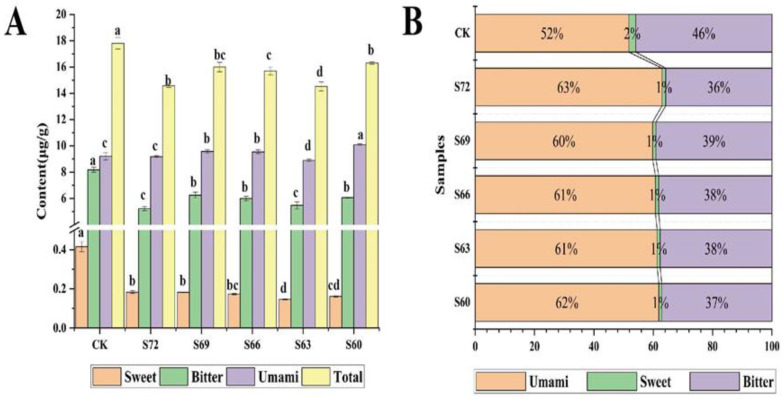
Amino acid content of black tea with different degrees of sun withering. (**A**) The content of different kinds of amino acids. (**B**) The ratio of different kinds of amino acids to total amino acids. Different lowercase letters on the bars indicate significant differences between groups at *p* < 0.05 level.

**Figure 5 foods-12-02430-f005:**
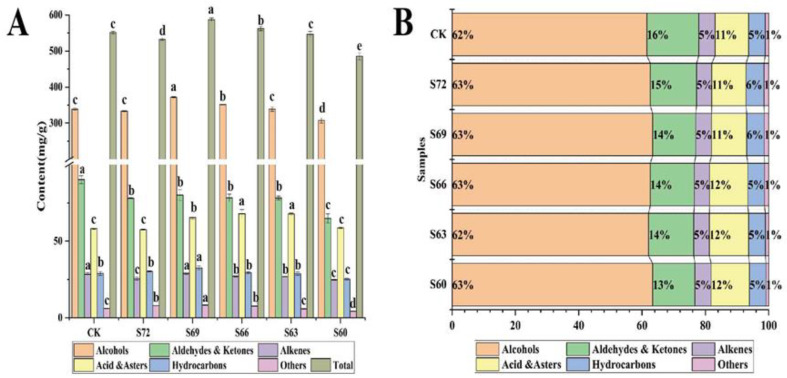
Analysis of volatile components in black tea with different degrees of sun withering. (**A**) Content plot of volatiles for each category. (**B**) The proportion of different kinds of volatiles to total volatiles. Different lowercase letters on the bars indicate significant differences between groups at *p* < 0.05 level.

**Figure 6 foods-12-02430-f006:**
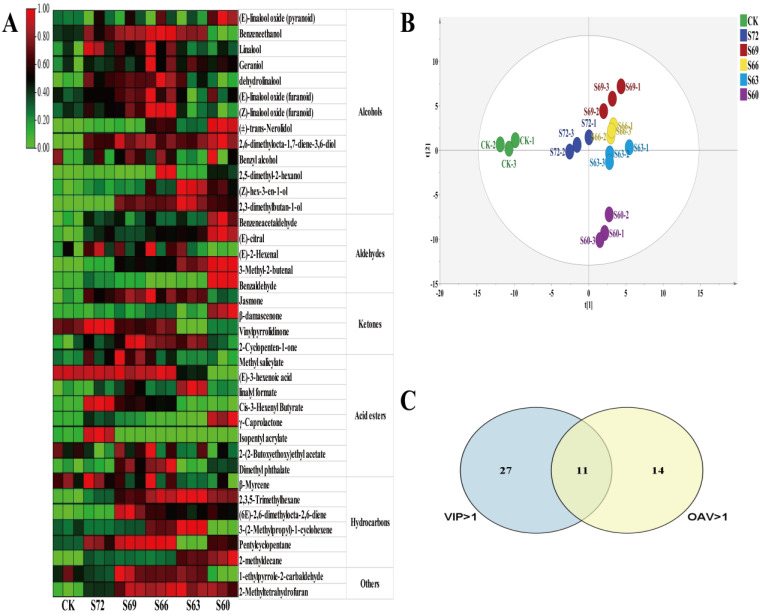
Analysis of volatile components in black tea with different degrees of sun withering. (**A**) Thermal plot of 38 differential compounds screened by OPLS-DA model with VIP > 1 and *p* < 0.05. (**B**) PCA plot (R^2^ = 0.0759, Q^2^ = 0.584). (**C**) VIP > 1 (*p* < 0.05) and OAV > 1 joint analysis Wayne plot.

**Figure 7 foods-12-02430-f007:**
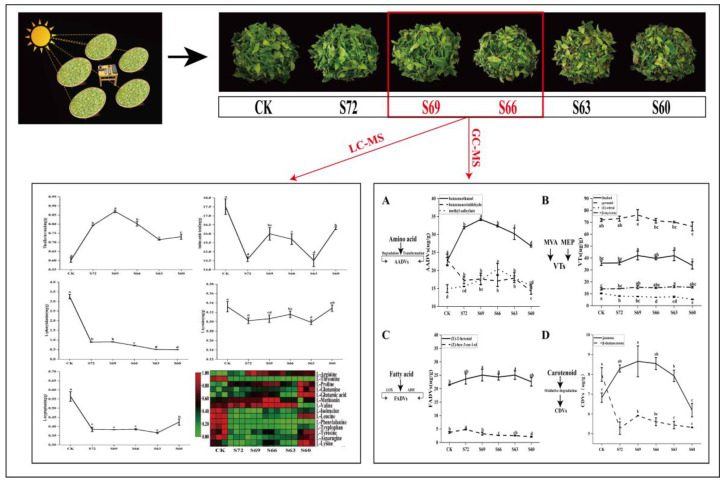
Conclusion diagram. (**A**) AADVs volatiles. (**B**) VTs volatiles. (**C**) FADVs volatiles. (**D**) CDVs volatiles. MVA: Mevalonate pathway. MEP: methylerythritol phosphate pathway. LOX: Lipoxygenase. ADH: Alcohol dehydrogenase. Different lowercase letters indicate significant differences at *p* < 0.05 level.

**Table 1 foods-12-02430-t001:** Analysis of OAV values of volatile substances of black tea with different degrees of sun withering (μg/g).

NO.	Compounds	Odor Threshold in Water (μg/L)	Odor Activity Values (OAVs)					
			CK	S72	S69	S66	S63	S60
1	Linalool	6 ^b^	426.57	429.24	503.57	474.72	502.43	403.98
2	Benzeneethanol	390 ^a^	1.51	2.13	2.29	2.17	2.02	1.79
3	Nerol	49 ^a^	2.24	1.92	1.86	1.86	1.84	1.47
4	Geraniol	7.5 ^a^	461.05	467.87	487.78	456.45	449.66	426.02
5	(Z)-hex-3-en-1-ol	13 ^a^	4.30	5.14	5.30	6.52	9.68	7.50
6	1-Octene-3-ol	1 ^a^	8.55	6.82	6.31	4.61	4.28	5.89
7	(E)-2-Hexenal	17 ^a^	264.11	290.65	308.15	299.34	308.32	277.75
8	Octanal	0.7 ^e^	15.37	30.55	103.42	103.53	112.51	37.06
9	Benzeneacetaldehyde	4 ^a^	5026.54	3782.8	3863.79	3741.69	3916.89	3119.73
10	(2E,6Z)-nona-2,6-dienal	0.03 ^b^	3723.37	1701.4	1578.46	1445.87	1603.33	1475.16
11	(2E)-2-Nonenal	0.19 ^a^	249.16	156.21	118.62	123.21	127.73	79.26
12	Decanal	3 ^c^	564.91	377.96	452.09	467.31	415.11	329.43
13	β-Cyclocitrala	3 ^a^	62.08	59.51	57.84	58.08	57.31	45.7
14	(2E)-2-Decenal	0.4 ^a^	95.86	102.09	79.69	69.5	56.38	75.83
15	(E)-citral	32 ^f^	7.35	5.77	5.52	5.01	5.34	3.79
16	(2E)-2-Octenal	4 ^c^	50.27	39.46	37.74	34.31	36.57	25.95
17	β-damascenone	0.002 ^b^	28,318.21	18,713.68	20,943.19	19,850.92	19,271.82	18,851.82
18	Jasmone	7 ^a^	143.29	174.2	181.89	179.53	166.49	130.03
19	β-ionone	8.4 ^a^	28.6	24.38	22.88	21.44	21.4	22.56
20	Hexanoic acid	35.6 ^d^	2.14	1.56	1.66	1.77	1.6	1.44
21	Methyl salicylate	40 ^a^	18.74	19.53	21.81	25.6	22.59	19.74
22	Dihydroactinidiolide	0.5 ^g^	414.37	345.2	377.67	369.67	349.58	377.31
23	δ-Decalactone	1.15 ^i^	135.31	151.32	274.94	308.56	426.99	152.21
24	β-Myrcene	15 ^a^	1.98	2	2.17	2.09	2.23	2.16
25	2,6,10,10-Tetramethyl-1-oxaspiro[4.5]dec-6-ene	0.2 ^h^	62.58	57.75	56.66	51.9	41.25	28.33

Note: OAV is calculated through dividing the concentration of a volatile compound by its odor threshold in water. Thresholds for different volatile substances are obtained from different references, with the small superscript letter for the cited literature. a: [41]; b: [42]; c: [43]; d: [44]; e: [45]; f: [46]; g: [47]; h: [48]; i: [49].

## Data Availability

Data is contained within the article.

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
