# Peer review of "Effects of Sun Withering Degree on Black Tea Quality Revealed via Non-Targeted Metabolomics"

_foods, 2023, doi:10.3390/foods12122430_

Round 1
Reviewer 1 Report
The manuscript need to be edited by English editors (native speaker) because there are many grammar mistake, wrong word that used and punctuations issues. I understand that English is the second language of the authors.

Author Response
Dear editor and reviewers:
We sincerely thank you and all the reviewers for the valuable comments and suggestions on how to improve the quality of our manuscript. We have carefully read the comments and revised the manuscript as suggested. Each response is shown in blue text, and changes in the manuscript are highlighted in red text.
Responses to Reviewers
To Reviewer #1:
Comments
Abstract:
- In this part the authors use many abbreviations such as SAFE-GC-MS and SPME-GC-MC, line 24 Ex. VIP>1 and 25 OAV>1. This should be in full at the first time in the manuscript especially in abstract part to make it more readable.
- Response :Thanks for the suggestion. We have checked the abstract and the whole manuscript and provided the full term for each abbreviation at its first mention, lines 22-23, lines 25-28, lines 399.
- The SPME (fiber) which type of SPME the authors use in this work such as DVB/C-WR/PDMS SPME fiber or what, so it is important to add these information.
- Response : Thanks for the suggestion. We have added information on the specific type of SPME in Materials and Methods, lines 147-148.
- Keywords: the word (Taste) should be deleted line 32.
- Response :Thanks for the suggestion, and we have removed the key word (Taste) as suggested, lines 35
Introduction
- In general, the introduction need to reorganized in systematic way in three paragraphs. The first one for black tea, second sun withering and third for metabolomics and the relationship between those factors.
- Response :Thank you very much for the insightful and constructive suggestion. We have divided the introduction into three paragraphs with a focus on black tea, sun withering, metabolomics and the relationship between those factors, lines 37-83.
- Lines 60 -70 More references need to add to give the readers clear idea about the background of this aroma or volatiles in the article.
- Response : Thanks for the suggestion, we have added more references about terpene volatile compounds in Introduction, lines 55-61.
- More information should add about VOCs that emitted from different cultivars of black tea or any plant in a family that is close to black tea plants.
- Response : Thanks for the suggestion, we have added information about VOCs (volatile compounds) of different varieties of black tea in Introduction, lines 61-65.
- The aim of this work is not clear, so the authors should add the scientific aim for this manuscript.
- Response : Thanks for the suggestion,and we have added the scientific aim of this study in Introduction, lines 78-83.
Materials and Methods
- How many (HS-SPME) were used in this work and how many replicates injection for each sample.
- Response : Thank you for the questions. A total of six samples were used for HS-SPME, with three replicates of each sample test and one injection for each replicate, lines 162-163.
- Did the authors do calibration for (GC-MS) before they used the instrument? They didn’t mentioned that.
- Response : Thank you for the questions. The GC-MS instrument is a commonly used testing instrument, which is maintained and calibrated regularly by the technical teachers in the State Key Laboratory of our university to ensure smooth experiment and data
- How many times the authors repeat the experiment.
- Response : Thank you for the question. The experiments were repeated three times for each sample, which has been mentioned in Materials and Methods, lines 162-163.
- All the information about the solvents and instruments are not mentioned correctly such as, An n-hexane (95%) was purchased from Sigma-Aldrich (catalogue number 270504-2L; Castle Hill, NSW, Australia). This should be mentioned in this way for whole Materials and Methods.
- Response : Thanks for the suggestion, and we have added the detailed information about the reagents in Table S1.
Results and Discussion
- In this part, the authors did not mentioned the identified volatiles as they mentioned in abstract part. Such as (65 non-volatile components were identified) (180 volatiles were identified). In contrast, there is just 25 volatiles were identified in table 1 , line 423. Where is the other VOC that identified ?. therefore, more tables need to be added.
- Response : Thank you for the questions and suggestion. We have added the data of non-volatile and volatile substances in supplementary materials.
- Many figures mentioned in this manuscript, I understand that the authors want to add all their results but not like this because it make the results and discussion more complex (complicated), so some of these figures must remove especially not necessary ones.
- Response : Thanks for the suggestion, and we have moved some of the figures to supplementary materials.
Conclusions
- In this part, the authors should give just a good findings because the is many repeated result for the results part and this is not a good way to write the conclusion. In this way, this part should reduce.
- Response : Thanks for the suggestion, and we have rewritten the conclusion section, lines 537-547.

Reviewer 2 Report
The manuscript entitled “Effects of sun withering degree on black tea quality revealed by non-targeted metabolomics” is devoted to the comprehensive investigation of processed tea samples with several modern analytical methods and sensory evaluation. The manuscript is well-organized and the conclusions are fully supported by the data.
I would recommend several things:
1) Denote SAFE method in the experimental section
2) Explain “infused leaf” parameter (taste of the infusion, when simple “taste” parameter stands for the tea?) and remove those +-0.00 values for it from the table. U may announce that all experts had the same results for this parameter, therefore statistics was not applicable.
3) Include PCA and OPLS-DA methods in 2.8 Data statistics and analysis.
4) It might be also recommended to enhance description of the earlier applied analytical protocols in the introduction section.
Minor editing of English language is preferable.
Author Response
Dear editor and reviewers:
We sincerely thank you and all the reviewers for the valuable comments and suggestions on how to improve the quality of our manuscript. We have carefully read the comments and revised the manuscript as suggested. Each response is shown in blue text, and changes in the manuscript are highlighted in red text.
Responses to Reviewers
To Reviewer #2:
Comments
Comments and Suggestions for Authors
- Denote SAFE method in the experimental section.
Response 1: Thanks for the suggestion, and we have added some information about SAFE, lines 152-162.
- Explain “infused leaf" parameter (taste of the infusion, when simple "taste" parameter stands forthe tea?) and remove those +-0.00 values for it from the table. U may announce that all expertshad the same results for this parameter, therefore statistics was not applicable.
Response 2: Thanks for the suggestion. Infused leaf is one of the tea sensory evaluation items to judge the tenderness, color, brightness, etc. of tea leaves. There is a correlation between infused leaf and tea quality, which is especially important when assessing tea quality. In addition, we have presented the value as “±0.00”, lines 216.
Chinese national standards < Methods of Sensory Evaluation of Tea > (GB/T 23776-2018).
- Include PCA and OPLS-DA methods in 2.8 Data statistics and analysis.
Response 3: Thanks for the suggestion, and we have added PCA and OPLS-DA in 2.8 Statistics and analysis of data, lines 167-168.
- lt might be also recommended to enhance description of the earlier applied analytical protocolsin the introduction section.
Response 4: Thanks for the suggestion, and we have added the earlier applied analytical protocolsin in Introduction, lines 67-72.